# The impact of autophagy on arbovirus infection of mosquito cells

**Doug E. Brackney** *, **Maria A. Correa, Duncan W. Cozens**

Center for Vector-Borne and Zoonotic Diseases, Department of Environmental Sciences, The Connecticut Agricultural Experiment Station, New Haven, Connecticut, United States of America

* doug.brackney@ct.gov

## Abstract

Macroautophagy is an evolutionarily conserved cellular process critical for maintaining cellular homeostasis. It can additionally function as an innate immune response to viral infection as has been demonstrated for a number of arthropod-borne (arbo-) viruses. Arboviruses are maintained in a transmission cycle between vertebrate hosts and invertebrate vectors yet the majority of studies assessing autophagy-arbovirus interactions have been limited to the mammalian host. Therefore we evaluated the role of autophagy during arbovirus infection of the invertebrate vector using the tractable Aag2 *Aedes aegypti* mosquito cell culture system. Our data demonstrates that autophagy is significantly induced in mosquito cells upon infection with two divergent arboviruses: dengue virus-2 (DENV-2; *Flaviviridae*, *Flavivirus*) and chikungunya virus (CHIKV; *Togaviridae*, *Alphavirus*). While assessing the role of autophagy during arbovirus infection, we observed a somewhat paradoxical outcome. Both induction and suppression of autophagy via torin-1 and spautin-1, respectively, resulted in increased viral titers for both viruses, yet suppression of autophagy-related genes had no effect. Interestingly, chemical modulators of autophagy had either no effect or opposite effects in another widely used mosquito cell line, C6/36 *Aedes albopictus* cells. Together, our data reveals a limited role for autophagy during arbovirus infection of mosquito cells. Further, our findings suggest that commonly used chemical modulators of autophagy alter mosquito cells in such a way as to promote viral replication; however, it is unclear if this occurs directly through autophagic manipulation or other means.

## Author summary

Arthropod-borne (arbo) viruses, specifically those transmitted by *Aedes aegypti* mosquitoes, cause significant morbidity and mortality and pose a continued public health threat worldwide. Many of these viruses lack vaccines or therapeutics and current mosquito control strategies are underperforming. For these reasons, identifying vulnerabilities within the transmission cycle that can be targeted will be critical to the development of novel control interventions. Autophagy is a highly conserved cellular pathway and previous studies manipulating this pathway have shown promise in minimizing viral infections in mammalian hosts. In this study we examined arbovirus-autophagy interactions within

**Data Availability Statement:** All relevant data are within the manuscript and its Supporting Information files.

**Funding:** This work was supported in part by grants from the Centers for Disease Control and

Prevention (https://www.cdc.gov/)(U01/
CK000509-01) (DEB) and the National Institute of
Health, National Institute of Allergy and Infectious
Diseases (https://www.niaid.nih.gov/)(AI099042)
(DEB and MAC). The funders had no role in the
study design, data collection and analysis, decision
to publish, or preparation of the manuscript.

**Competing interests:** The authors have declared
that no competing interests exist.

mosquito cells. The goal was to elucidate the role of autophagy during infection of these
cells in hopes of identifying critical interactions that can be targeted by novel approaches
to block infection of and transmission by vector mosquitoes.

## Introduction

Arthropod-borne (arbo) viruses, specifically those of the families *Flaviviridae* and *Togaviridae*,
cause significant morbidity and mortality and pose a continued public health threat world-
wide. This is highlighted by the sustained transmission of dengue virus (DENV; *Flaviviridae*,
*Flavivirus*) throughout the tropics, the emergent epidemics of Zika virus (ZIKV; *Flaviviridae*,
*Flavivirus*), chikungunya virus (CHIKV; *Togaviridae*, *Alphavirus*) and West Nile virus (WNV;
*Flaviviridae*, *Flavivirus*) in the Americas, and the sporadic outbreaks of yellow fever virus
(YFV; *Flaviviridae*, *Flavivirus*) in Africa and South America. Arboviruses are maintained in a
transmission cycle between vertebrate hosts and invertebrate vectors. Because therapeutics
and vaccines are lacking for many of these viruses, public education and vector control strate-
gies are the only defenses available to minimize disease burden and epidemic risk. However, as
insecticide resistance is becoming more commonplace it is critical that novel approaches for
preventing and treating arboviral diseases, targeting either vector or host, be developed.

Macroautophagy (herein referred to as autophagy) is an essential cellular process required
for maintaining homeostasis and plays a crucial role in development, cell differentiation and
immunity [1, 2]. Consequently, abnormal autophagic activity has been linked to a number of
pathologies, including cancer and neurodegenerative diseases [3]. This evolutionarily con-
served pathway mediates the degradation and recycling of cellular components. Upon induc-
tion, protein aggregates, damaged or dysfunctional organelles and foreign bodies are
sequestered into cup-shaped double-membrane vesicles termed phagophores or isolation
membranes [4]. Appropriation of cargo can occur in a non-selective manner known as bulk
autophagy whereby cytoplasmic material is indiscriminately engulfed, or through selective
autophagy, which specifically targets poly-ubiquitin tagged proteins [5]. Through a series of
kinase signaling cascades and recruitment events, the phagophore elongates, eventually
encompassing the cargo and forming an autophagosome [4]. Subsequently, the mature autop-
hagosomes fuse with lysosomes resulting in the acidification of the newly formed autophagoly-
sosome and degradation of the cargo [4]. Targeting aspects of the autophagy pathway has
become a promising strategy for combatting a number of diseases [6, 7].

Numerous stimuli have the ability to activate autophagy including innate immunity and
cellular stress. It is therefore not surprising that viral infections often induce an autophagic
response. In fact, studies have demonstrated that both flaviviruses and alphaviruses induce
autophagy during infection and that this can occur through a number of mechanisms, all of
which culminate in the inhibition of the Akt-TOR signaling pathway [8–14]. During alpha-
virus infection, autophagy functions as an antiviral defense. Previous studies have demon-
strated that activation of the autophagy pathway via recombinant Beclin 1 reduces Sindbis
virus and CHIKV titers and improves clinical outcomes in mice. This process is mediated by
the autophagy cargo receptor p62, which recognizes poly-ubiquitin tagged viral capsid proteins
and transports them to maturing autophagosomes for degradation [10, 13, 15–18]. In compar-
ison, flavivirus-autophagy interactions are much more nuanced [19]. For instance, during
DENV-2 infection, p62 actively targets DENV-2 proteins for degradation and replication com-
plexes associated with the ER (endoplasmic reticulum) are targeted by reticulophagy; however,
DENV-2 benefits from an energetically favorable environment as a result of virus-induced

lipophagy and efficient processing of mature virions [9, 19–21]. While it is clear that arboviruses closely interact with autophagy during infection, these studies have been limited to either non-vector organisms or mammals. As a result we know very little about the role of autophagy during arbovirus infection within the invertebrate vector [22].

To address this knowledge gap we investigated the role of autophagy during DENV-2 and CHIKV infection of Aag2 mosquito cells derived from the primary vector *Aedes aegypti*. Using an anti-Atg8 (mammalian homolog LC3) specific antibody and recombinant Atg8-EGFP provided *in trans*, we assessed autophagy induction and flux by immunoblot and microscopy. We observed an increase in the accumulation of Atg8 positive puncta and increased conversion of Atg8 to Atg8-PE (PE; phosphatidylethanolamine). Further, we found that chemical inhibition and induction of autophagy both increased viral titers, while suppression of autophagy related genes had no effect. Interestingly, chemical modulators had minimal effects on DENV-2 titers in another commonly used mosquito cell line, *Aedes albopictus* C6/36, and spautin-1 inhibition of autophagy significantly decreased DENV-2 titers in mammalian cells as previously reported [21]. Together, these data reveal a limited role for autophagy during DENV-2 and CHIKV infection of mosquito cells and highlights differences in autophagy-virus interactions between cell culture systems. Further, our data suggest that outcomes associated with commonly used chemical modulators of autophagy are cell-dependent and may result from cell-specific interactions with the chemicals.

## Materials and methods

### Cell lines & virus strains

Autophagy was modeled in three different cell lines, the mosquito-derived *Aedes albopictus* C6/36 (ATCC; American Type Culture Collection) and *Aedes aegypti* Aag2 cells (Generously provided by Dr. Gregory Ebel, Colorado State University) and mammalian cell line BHK-21 clone 15 (Syrian golden hamster kidney cells) (Generously provided by Dr. Rushika Perrera, Colorado State University). The two mosquito cell lines were maintained at 28°C in the presence of $CO_2$, and the BHK cells were maintained at 37°C with $CO_2$. All cells were grown in media containing 10% fetal bovine serum, sodium bicarbonate, 100 U/ml penicillin, 100 μg/ml streptomycin, 0.25 μg/ml amphotericin B, L-glutamine, and non-essential amino acids. This media was used in all transfections, infections and plaque assays. Cell infections were carried out using viruses from two major arbovirus families, *Flaviviridae* and *Togaviridae*. DENV-2 strain Jam 1409 (Jamaica) was passaged on C6/36 cells and CHIKV strain R99659 (Caribbean lineage, British Virgin Islands, 2013) was passaged on Vero E6 cells (African green monkey kidney cells).

### Pharmacological modulators

To assess the effect of pharmacological modulators on autophagy, the following drugs were used in cell culture infections of C6/36, Aag2, and BHK21 cells: torin-1, an autophagy inducer that works early in the autophagy pathway by inhibiting mTOR kinase, and spautin-1 (specific and potent autophagy inhibitor 1) (Tocris Bioscience), which inhibits autophagy by interfering with USP10 and USP13, two ubiquitin-specific peptidases, resulting in the degradation of class III phosphatidylinositol-3 kinase complexes during the initial steps of autophagy [23]. Bafilomycin A1 (InvivoGen), an inhibitor of V-ATPase, was also used in inhibition of late stage autophagy in infectious and non-infectious immunoblotting of Aag2 cells. A further discussion of these chemicals is not included as others have nicely summarized their specific activity within the autophagy pathway [24]. All drugs were prepared in dimethyl sulfoxide (DMSO) and stored at -20°C.

## Immunoblots

Two days after Aag2 cells were seeded at $1.5 \times 10^6$ cells/well in a 12-well plate, cells were treated with 1% DMSO, 1 μM bafilomycin A1, 1 μM torin-1 or 10 μM spautin-1. Cells were harvested 24 hours later using RIPA buffer. Total protein in cell lysates was quantified using a Pierce BCA Protein Assay kit (ThermoFisher) on a microplate reader. An equal amount of protein from each sample was separated on a 16% Tris-glycine gel and transferred to a polyvinylidene difluoride (PVDF) membrane. Autophagy was detected using a rabbit anti-Atg8 antibody at a 1:8,000 dilution in PBS + 5% BSA + 1% Tween. Polyclonal anti-Atg8 antibody was generated by ProSci Inc. as previously described [25]. The blot was then probed with horseradish peroxidase (HRP) goat anti-rabbit antibody at a 1:20,000 dilution. Actin was detected using a rabbit anti-β-Actin antibody (Abcam) in PBS + 5% BSA + 1% Tween at a 1:8,000 dilution. The secondary probe was performed using the same secondary antibody as above. Membranes were developed in SuperSignal West Pico chemiluminescent substrate (Thermo Fisher) for 5 minutes at room temperature. Band intensities were quantified using ImageJ [26].

The effect of virus infection in combination with pharmacological modulators on autophagy was also assayed via western blot. Aag2 cells were prepared as above and infected with DENV-2 or CHIKV at a multiplicity of infection (M.O.I.) of 0.5. After one hour of infection at 28˚C, DMSO, 1 μM bafilomycin A1, 1 μM torin-1, or 10 μM spautin-1 were added to the wells. For DENV-2, cells were harvested 48 hours post infection (hpi) using RIPA buffer and tested for Atg8 expression as above. Due to a difference in viral replication kinetics, CHIKV samples were harvested at 24 hpi. Band intensities of Atg8-PE and β-actin were determined by ImageJ. Autophagy induction was quantified by calculating the ratio of Atg8-PE to β-actin for each sample. The fold change in Atg-PE induction for each treatment was determined by comparing it to the DMSO control. All experiments were completed in triplicate.

## Autophagy gene suppression

*Aedes aegypti* Atg5, Atg14, Atg8 and luciferase genes were amplified using primer sets containing the T7 promotor (Table 1). Amplicons were PCR purified and a T7 Megascript kit was used to synthesize dsRNA molecules for transfection per the manufacturer's recommendations (Ambion). Two days prior to transfection, Aag2 cells were seeded at $1.5 \times 10^6$ cells/well in 12-well plates. 1 μg/well of dsRNA was combined with OPTI-MEM and Lipofectamine 2000 transfection reagent (Invitrogen) and allowed to incubate at room temperature for 20 minutes.

**Table 1. Primer list.**

| Gene | Accession # | Primer | Sequence (5'-3') |
|---|---|---|---|
| AaeAtg5 | XM_001661191 | T7-Forward | TAA TAC GAC TCA CTA TAG GGT CCG ATG AAA CCG ATG TC |
| | | T7-Reverse | TAA TAC GAC TCA CTA TAG GGC ATC GAA CTG AAA TGT G |
| | | RT-qPCR-Forward | AGT TCG ATG TTA TGC CGA GG |
| | | RT-qPCR-Reverse | GAT AGC TGA GGT GTT CCG AG |
| AaeAtg14 | XM_001652367 | T7-Forward | TAA TAC GAC TCA CTA TAG GGT TTC TCG AGC AGT GAC GGA |
| | | T7-Reverse | TAA TAC GAC TCA CTA TAG GCG CAA TTC GTT CAA CTG GCA |
| | | RT-qPCR-Forward | ACT CGA AAC GGC ACT TCC AT |
| | | RT-qPCR-Reverse | TGT GGC GTA GTT GCT TCT CG |
| AaeAtg8 | XM_001652521 | T7-Forward | TAA TAC GAC TCA CTA TAG GAT GAA ATT TCA ATA CAA G |
| | | T7-Reverse | TAA TAC GAC TCA CTA TAG GAC TTG TTT CCA TAC ACA TTC |
| AaeGAPDH | XM_011494724 | RT-qPCR-Forward | ACG TGA ACA GAC GCT AGT TAT |
| | | RT-qPCR-Reverse | GTG GGT CGA ATC GTA CTT GAA |

The media on the Aag2 cells was discarded and replaced with 1 mL of Opti-MEM. The lipid-nucleic complexes were then added to the wells containing OPTI-MEM and were incubated at 28°C for four hours, after which the media was discarded and again replaced with the standard 10% FBS media. Cells were harvested 48 hours after transfection by manually scraping the cell monolayer using a 1 ml pipette tip. The cells were pelleted and re-suspended in QVL Lysis buffer from an Omega Bio-tek Viral RNA Extraction kit or RIPA buffer. Total RNA was then extracted following the manufacturer's instructions. Extractions were DNase treated and purified via phenol/chloroform extraction.

Silencing efficiency of Atg5 and Atg14 was determined through RT-qPCR with Universal SYBR Green master mix (Bio-Rad), using luciferase samples as the non-targeting control group and GAPDH as a reference gene (Table 1). Silencing efficiency of Atg8 was determined by Western blotting. Two days after dsAtg8 was transfected into Aag2 cells, cells were infected with DENV-2 as above. Cells were harvested in RIPA buffer 24 hpi. Immunoblotting was performed as previously described.

## Virus titration

Two days after Aag2 cells were seeded at $1.5 \times 10^6$ cells/well in a 12-well plate, cells were infected with DENV-2 or CHIKV at an M.O.I. of 0.1. After infection for one hour at 28°C, cells were treated in duplicate with 1% DMSO, 1 µM bafilomycin A1, 1 µM torin-1 or 10 µM spautin-1. Virus supernatants were collected 24 and 48 hours after drug application. Finally, Aag2 cells were seeded and transfected with dsLuciferase, dsAtg5, dsAtg8, or dsAtg14 as previously described. Two days after transfection, the cells were infected with DENV-2 or CHIKV and virus supernatants were harvested at 24 and 48 hpi.

The effect of cell line on DENV-2 virus titers was determined by following the above protocol in a mammalian cell line (BHK21 c15) and a mosquito cell line with a dysfunctional RNA interference pathway (C6/36) [27, 28]. C6/36 cells were seeded at $1.5 \times 10^6$ cells/well and BHK cells were seeded at $5 \times 10^5$ cells/ well in 12 well plates. The following day, both cell lines were infected with DENV-2 and treated with 1% DMSO, 1 µM torin-1 or 10 µM spautin-1. Titers were harvested at 48 hpi. Each experiment was completed in triplicate with two biological replicates per experimental replicate.

For titration of DENV-2, BHK21 c15 cells were grown to a confluent monolayer in 12-well plates and infected with 10-fold serial dilutions of virus for 1 hour at room temperature and overlaid with a mixture of the standard media and 5% methyl cellulose. After incubation at 37°C for four days, the cells were fixed with 7.4% formaldehyde and stained with gentian violet. Viral titers were determined by counting plaques. For titration of CHIKV, the above protocol was followed in Vero cells.

## Confocal microscopy

In order to track autophagy during virus infection we quantified Atg8+ puncta by confocal microscopy. While we possess a polyclonal anti-Atg8 antibody, it did not produce reliable and consistent results. Therefore, we generated an expression vector to provide EGFP tagged Atg8 *in trans*. This was achieved by cloning the full length *Ae. aegypti* Atg8 gene (AAEL007162) into the pIEx-EGFP vector 3' to EGFP by Gibson Assembly. The generation of pIEx-EGFP has been described previously [28]. Aag2 cells were seeded on coverslips at a density of $1.5 \times 10^6$ cells/well in a 12 well plate and 24 hrs. later transfected with 1.5 µg/ well of pIEx-EGFP-Atg8 using Lipofectamine 2000 transfection reagent according to the manufacturer's instructions. Two days post-transfection, cells were infected (M.O.I. 0.1) and/ or chemically treated. Cells were fixed in 4% paraformaldehyde 24 hrs. post-treatment, washed 3x in warm PBS and

mounted on microscope slides using ProLong Gold + DAPI. Microscopy was performed on a Leica SP5 confocal microscope at 100x magnification with a 3x zoom. Because the efficiency of transfection is relatively low in Aag2 cells, images were captured by scanning the slide and identifying field of views that contained multiple cells with EGFP expression. Five images per sample per experiment were acquired. Approximately 50 cells were imaged per treatment. Experiments were completed in triplicate and all identifying information was blinded to the microscopist. Quantification of puncta per cell was completed using the ImageJ software with the Puncta Analyzer plug-in. It should be noted that infection status of each cell could not be verified at the time of microscopy due to the rate of virus replication in these cells. Samples were harvested 24 hpi, yet viral antigen is in low abundance at this early time point making it difficult to detect signal above background auto-fluorescence. This limitation made it difficult to correlate virus replication with Atg8+ puncta accumulation.

### Statistical analysis

Immunoblot fold-change differences and puncta abundance were analyzed by one-way ANOVA with a Sidak's multiple comparisons test ($\alpha = 0.05$). All viral titer data was analyzed by one-way ANOVA with a Dunnett's multiple comparisons test ($\alpha = 0.05$). Suppression of Atg5 and Atg14 was calculated with a two-tailed t-test ($\alpha = 0.05$). Grubb's test for outliers ($\alpha = 0.05$) was used to determine if specific data points should be excluded from analysis. Data was analyzed using GraphPad Prism 6.

## Results

### Autophagy in mosquito cells

Because autophagy has been understudied in mosquitoes it is unclear if traditional methods for manipulating and analyzing autophagy, such as the use of chemical compounds, will be effective and/ or toxic in mosquito cells. Consequently, we verified the ability of commonly used chemical compounds to modulate autophagy in Aag2 cells. Treatment of Aag2 cells for 24 hrs. with bafilomycin A1 (inhibits autophagosome/ lysosome fusion), torin-1 (induces autophagy) and spautin-1 (blocks autophagy induction) alone had no discernible effects on the ratio of Atg8-PE to β-actin compared to the DMSO control cells as determined by immunoblot (Fig 1). However, the combination of torin-1 and bafilomycin A1 resulted in a significant increase in Atg8-PE to β-actin ratios. By inhibiting autophagolysosome acidification with bafilomycin A1 we were able to observe signs of autophagic activity suggesting that Aag2 cells have a high rate of autophagic flux (the steady state rate of autophagosomal turnover) and providing evidence that torin-1 and bafilomycin A1 are functioning as expected with regards to autophagy. As a result of the high autophagic flux, the addition of spautin-1 had no effect on baseline autophagy levels. However, when added to cells treated with torin-1 and bafilomycin A1, spautin-1 reduced the ratio of Atg8-PE to β-actin thereby validating its ability to inhibit autophagy in Aag2 cells (Fig 1). The MTT assay for cell viability revealed that none of these compounds displayed statistically significant levels of toxicity to Aag2 cells at the concentrations used in these studies (S1 Fig).

### Arbovirus induced autophagy

Numerous studies have demonstrated that arboviruses induce autophagy upon infection of mammalian cells [12, 16, 17, 19–21]. To examine if autophagy is similarly induced upon arbovirus infection in mosquito cells, Aag2 cells were infected with DENV-2 (48 hpi) and CHIKV (24 hpi) and autophagic activity quantified. Infection with either DENV-2 or CHIKV alone

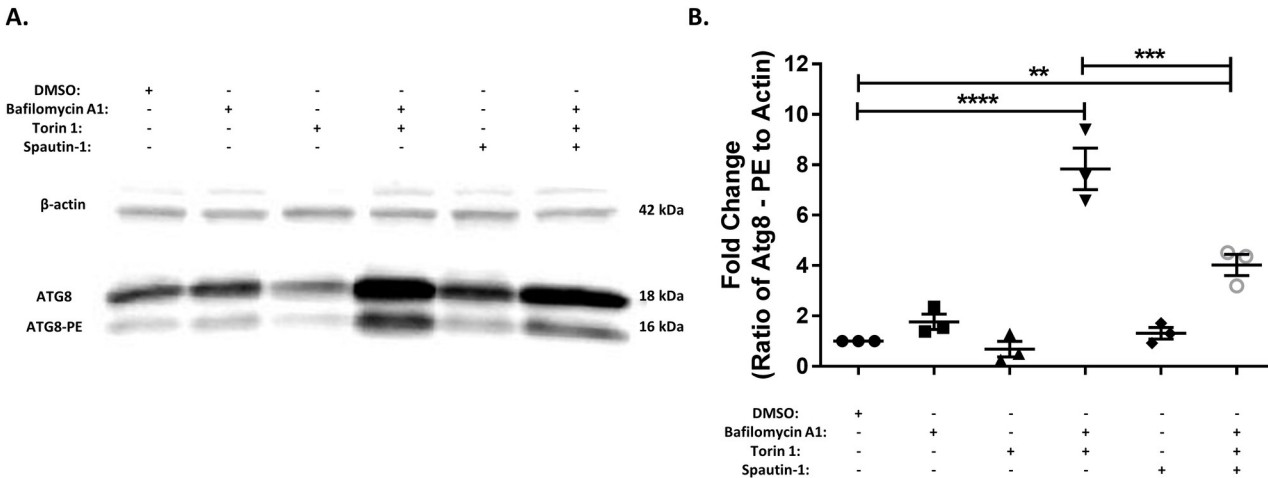

**Fig 1. Common pharmacological manipulators of autophagy function as expected in mosquito cells.** A) Representative immunoblot of Aag2 lysates upon chemical treatment (1% DMSO, 1 μM bafilomycin A1, 1 μM torin-1 or 10 μM spautin-1) 24 hpi. B) Fold-change in the ratio of Atg8-PE to β-actin band intensities as determined by ImageJ. Includes data from three experimental replicates. Data were analyzed by one-way ANOVA with a Sidak's multiple comparisons test. (*) $p < 0.05$, (**) $p < 0.01$, (***) $p < 0.001$.

had no effect on the ratio of Atg8-PE to β-actin as determined by immunoblot (Figs 2A, 2B, 3A and 3B). Similar results were observed for another flavivirus, Zika virus (ZIKV) at 48 hpi (S2A and S2B Fig). As previously determined, our inability to observe autophagy induction may be due to the high rate of autophagic flux associated with these cells. Consequently, addition of bafilomycin A1 to the DENV-2 infections significantly increased Atg8-PE to β-actin ratios (Fig 2A and 2B). Interestingly this did not increase ratios during CHIKV or ZIKV infection compared to the bafilomycin A1 control group (Fig 3A and 3B)(S2A and S2B Fig). To further validate these findings, Aag2 cells were transfected with a plasmid constitutively expressing Atg8-EGFP and infected in the presence or absence of bafilomycin A1. As with the immunoblot, DENV-2 infections (48 hpi) in the absence of bafilomycin A1 had no effect on autophagic activity, while the addition of bafilomycin A1 significantly increased Atg8+ puncta/ cell compared to the controls (Fig 1C and 1D). Similarly, increases in Atg8+ puncta/ cell were observed during CHIKV (24 hpi) and ZIKV (48 hpi) infection in the presence of bafilomysin A1 (Fig 3C and 3D) (S2C and S2D Fig). Nevertheless, the totality of the microscopy and immunoblot data provide confidence that, like mammalian systems, arboviruses induce autophagy in mosquito cells.

## Autophagy arbovirus interactions

Autophagy plays a central role in both innate and adaptive immunity [1]. Numerous studies have demonstrated that autophagy can be antiviral; however, some viruses have evolved mechanisms to either circumvent autophagy or usurp components of the pathway for their own benefit [29]. To ascertain the role of autophagy during arbovirus infection of mosquitoes, the impacts of pharmacological modulators of autophagy on arbovirus replication in Aag2 mosquito cells was assessed. If autophagy were functioning in an antiviral manner, inhibition of autophagy would result in increased viral titers and induction would result in decreased titers. Indeed the application of spautin-1 resulted in significantly increased DENV-2 titers at 24 and 48 hpi (Fig 4A) and CHIKV titers at 24 and 48 hpi (Fig 4B). Similar results were observed for ZIKV at 48 hpi (S3 Fig). Interestingly, induction of autophagy with torin-1 also resulted in

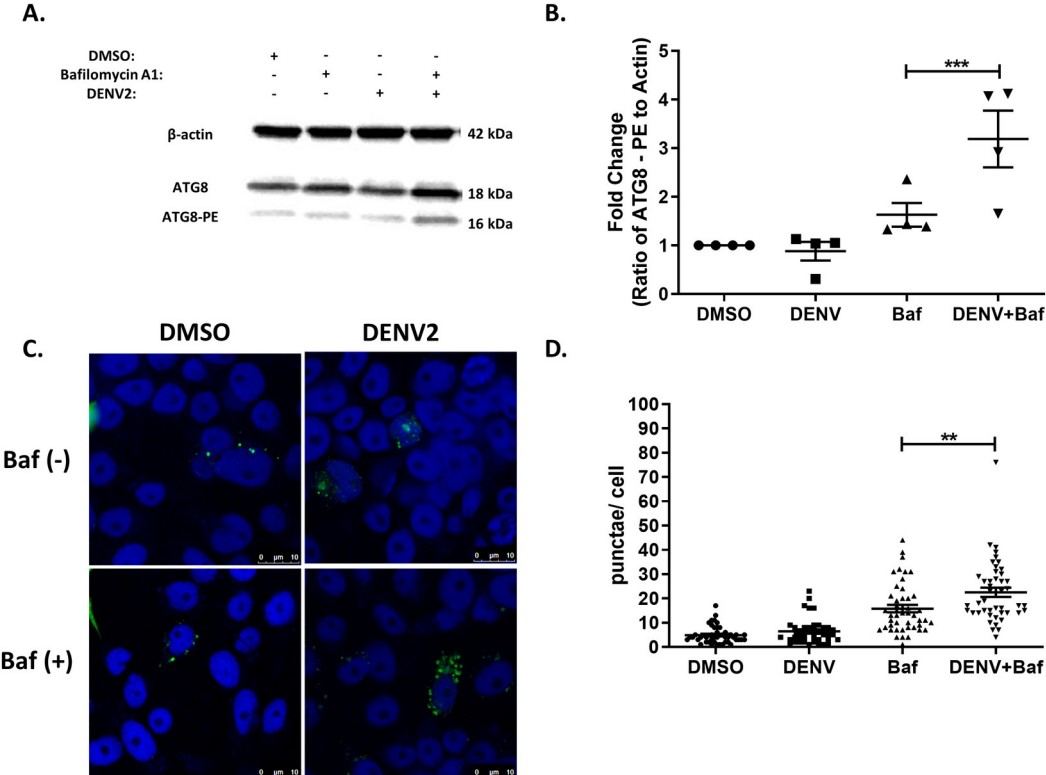

**Fig 2. DENV-2 induces autophagy in *Aedes aegypti* Aag2 cells.** A) Representative immunoblot of Aag2 lysates upon DENV-2 infection (M.O.I. 0.5) and/ or chemical treatment (1% DMSO, 1 μM bafilomycin A1, 1 μM torin-1 or 10 μM spautin-1) 48 hpi. B) Fold-change in the ratio of Atg8-PE to β-actin band intensities as determined by ImageJ. Includes data from four experimental replicates. C) Representative confocal microscopy images of Atg8-EGFP expressing Aag2 cells ± DENV-2 infection (M.O.I. 0.1) and ±1 μM bafilomycin A1 24 hpi. Blue (nuclei), green (Atg8-EGFP + puncta). D) The number of Atg8+ puncta were quantified using the ImageJ Puncta Analyzer plug-in from ~50 Atg8-EGFP expressing Aag2 cells ± DENV-2 infection (M.O.I. 0.1) and ±1 μM bafilomycin A1 24 hpi. Combined data from three blinded experimental replicates. Data were analyzed by one-way ANOVA with a Sidak's multiple comparisons test. (*) p<0.05, (**) p<0.01, (***) p<0.001.

significantly increased titers for all three viruses at all time points (Fig 4A and 4B) (S3 Fig). Further, blockage of V-ATPase activity with bafilomycin A1 resulted in an increase in autophagolysosomal pH levels and significantly increased DENV-2 and ZIKV titers at all time points (Fig 4A) (S3 Fig), but not those of CHIKV (Fig 4B). Due to the unexpected and paradoxical findings, we examined the effects of autophagy through suppression of three genes within the autophagy pathway: Atg5 (required for phagophore elongation), Atg8 (required for phagophore elongation and membrane curvature) and Atg14 (essential autophagy regulator). Treatment of Aag2 cells with gene specific dsRNA resulted in significant suppression of Atg5 and Atg14 as determined by RT-qPCR as well as depletion of Atg8 as determined by immunoblot 72 hpt (hours post treatment) (S4 Fig). Suppression of the autophagy related genes had no discernible effect on DENV-2 titers 24 or 48 hpi compared to the control (Fig 5A). Likewise suppression of Atg5 and Atg14 had no effect on CHIKV at 24 or 48 hpi or ZIKV titers at 48 hpi (Fig 5B)(S3 Fig). Together these results suggest that chemical modulators of autophagy significantly alter arbovirus replication in Aag2 cells, yet it is unclear if this is due to modulation of the autophagy pathway or through an as of yet unidentified non-autophagy mediated mechanisms.

## Autophagy response to arboviruses in other systems

Due to the unexpected results associated with chemical treatment in Aag2 cells, we examined if the observed outcomes were consistent in other host systems. Others have previously shown that spautin-1 mediated inhibition of autophagy in mammalian cells, specifically BHK-21, drastically reduces DENV-2 titers [21]. In an effort to recapitulate these findings we treated DENV-2 infected BHK-21 cells with spautin-1 and observed a significant reduction in DENV-2 titers 48 hpi (Fig 6). Induction of autophagy with torin-1 had no effect (Fig 6). Next we wanted to determine if our results in Aag2 cells were similar in another commonly used mosquito cell line, C6/36 *Aedes albopictus* cells. As before, treatment of Aag2 cells with both torin-1 and spautin-1 resulted in significantly increased DENV-2 titers 48 hpi; however, we observed a significant decrease in titers upon torin-1 treatment and no effect upon spautin-1 treatment in C6/36 cells (Fig 7). This data suggests that these commonly used chemical compounds can have profoundly different effects on host cell-virus interactions.

## Discussion

Manipulation of the autophagy pathway has become a promising approach for treating a number of diseases including viral infections [6, 7, 18]. Because arboviruses are maintained in a

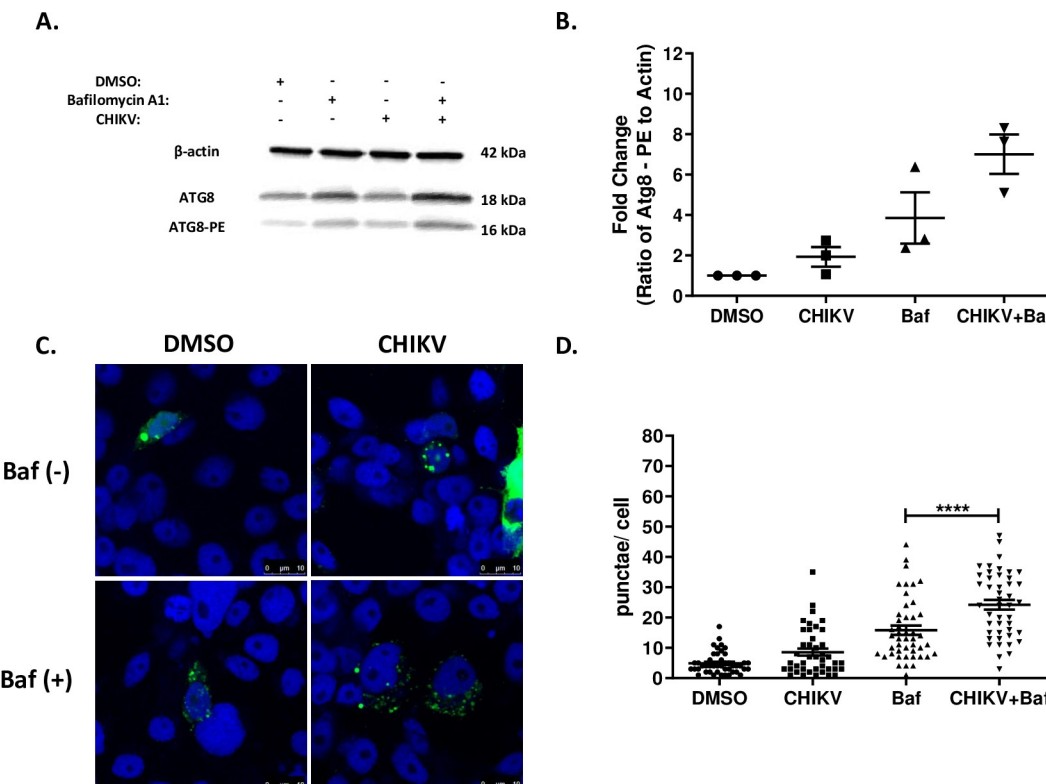

**Fig 3. CHIKV induces autophagy in *Aedes aegypti* Aag2 cells.** A) Representative immunoblot of Aag2 lysates upon CHIKV infection (M.O.I. 0.5) and/ or chemical treatment (1% DMSO, 1 μM bafilomycin A1, 1 μM torin-1 or 10 μM spautin-1) 24 hpi. B) Fold-change in the ratio of Atg8-PE to β-actin band intensities as determined by ImageJ. Includes data from four experimental replicates. C) Representative confocal microscopy images of Atg8-EGFP expressing Aag2 cells ± CHIKV infection (M.O.I. 0.1) and ±1 μM bafilomycin A1 24 hpi. Blue (nuclei), green (Atg8-EGFP + puncta). D) The number of Atg8+ puncta were quantified using the ImageJ Puncta Analyzer plug-in from ~50 Atg8-EGFP expressing Aag2 cells ± CHIKV infection (M.O.I. 0.1) and ±1 μM bafilomycin A1 24 hpi. Combined data from three blinded experimental replicates. Data were analyzed by one-way ANOVA with a Sidak's multiple comparisons test. (*) p<0.05, (**) p<0.01, (***) p<0.001.

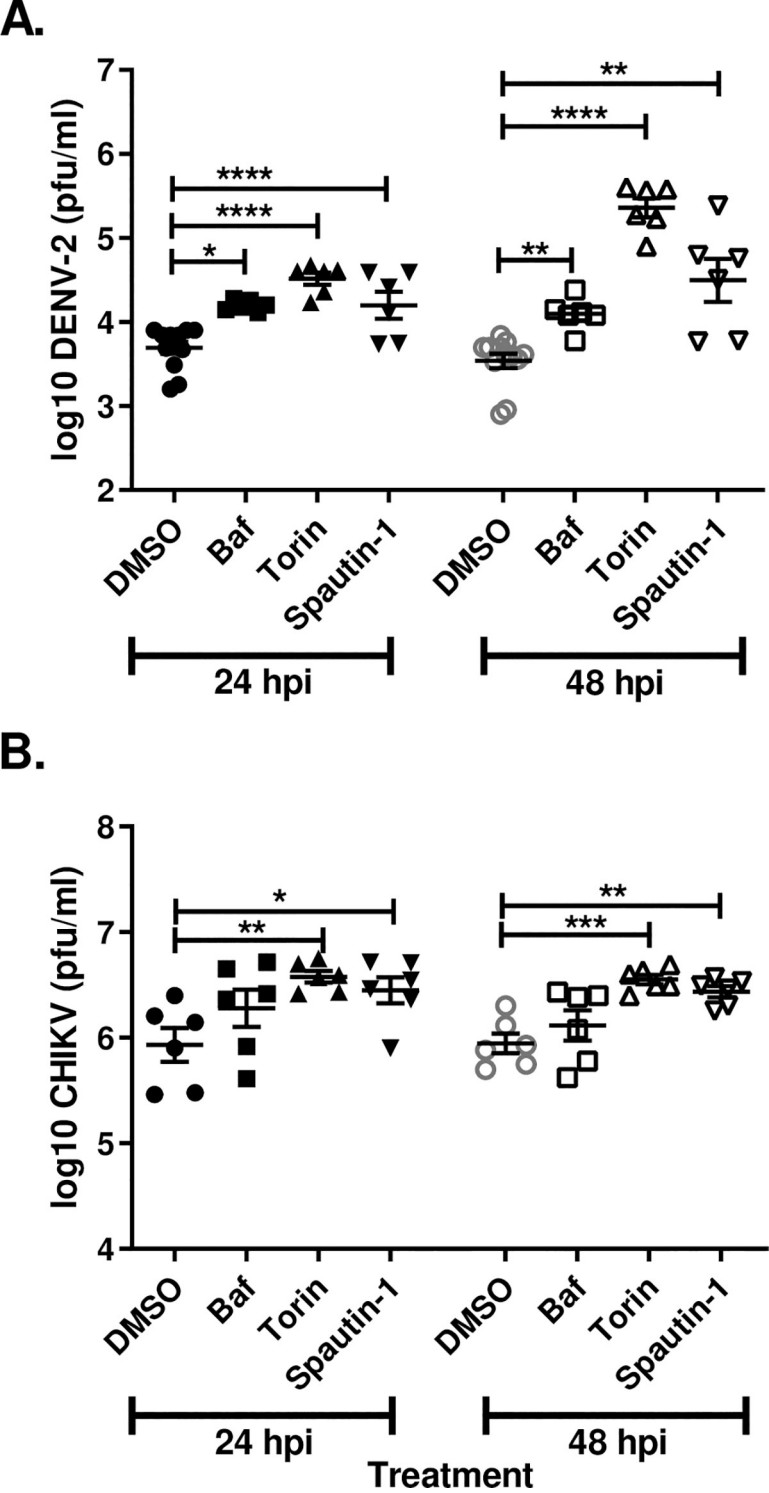

**Fig 4. Pharmacological modulators of autophagy uniformly increase DENV-2 and CHIKV titers in Aag2 cells.**
*Aedes aegypti* Aag2 cells were infected with A) DENV-2 or B) CHIKV followed by chemical treatment (1% DMSO, 1 μM bafilomycin A1, 1 μM torin-1 or 10 μM spautin-1). Samples were collected for titration 24 and 48 hpi for DENV-2 and CHIKV. Data was analyzed by one-way ANOVA with a Dunnett's multiple comparisons test. (*) p<0.05, (**) p<0.01, (***) p<0.001.

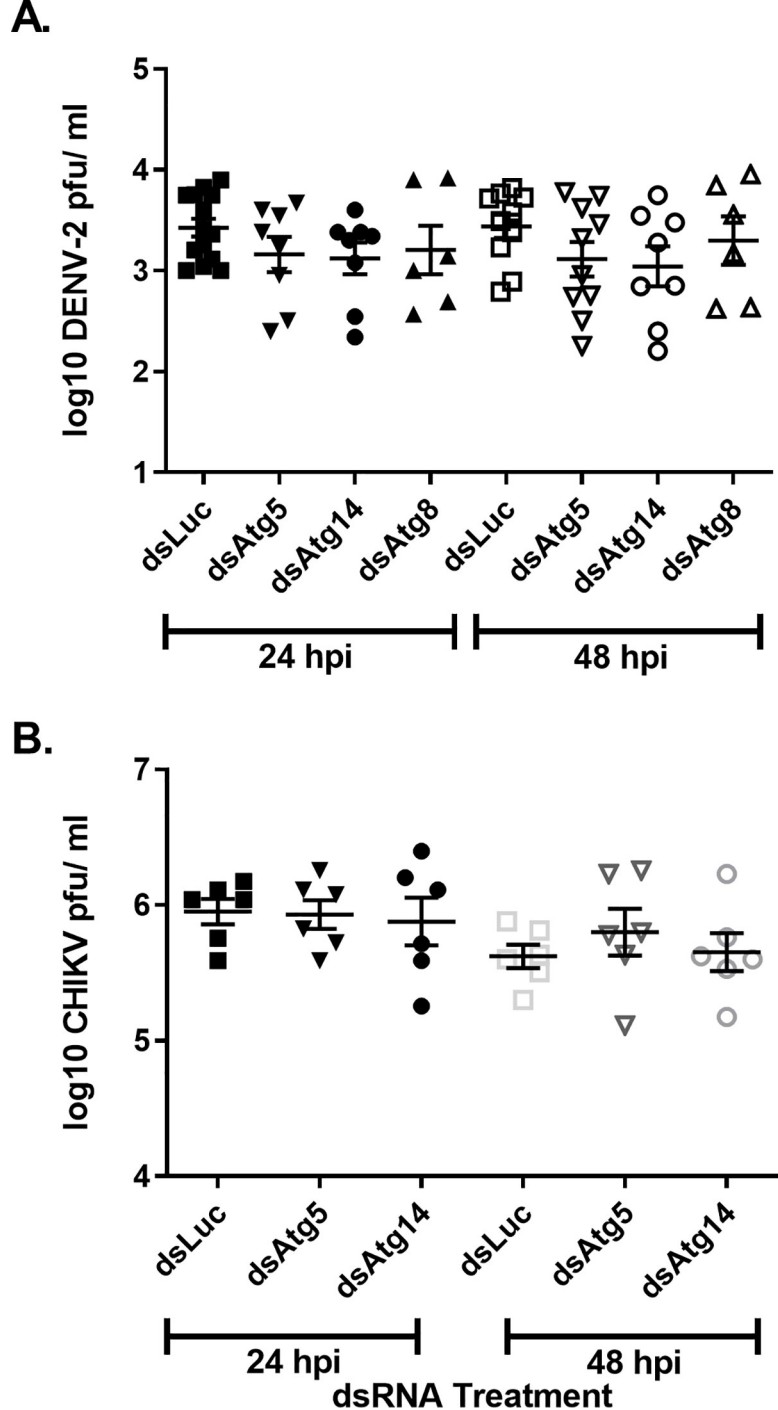

**Fig 5. Suppression of autophagy related genes has no effect on DENV-2 or CHIKV titers in Aag2 cells.** *Aedes aegypti* Aag2 cells were treated with dsRNA against Atg5, Atg14, Atg8, and non-specific control luciferase genes two days prior to infection with either A) DENV-2 or B) CHIKV. Samples were collected for titration 24 and 48 hpi. Data was analyzed by one-way ANOVA with a Dunnett's multiple comparisons test. (*) $p < 0.05$, (**) $p < 0.01$, (***) $p < 0.001$.

two-host system, the invertebrate vector and vertebrate host, there exists the possibility that autophagy can be manipulated in one or both systems in order to reduce disease burden and/ or transmission. Despite this two-host dependency, our understanding of the autophagy-

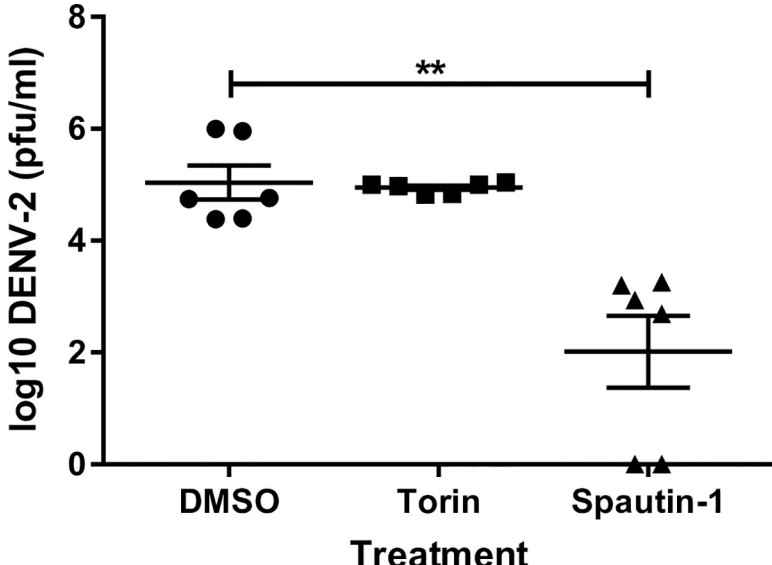

**Fig 6. Pharmacological inhibition of autophagy decreases DENV-2 titers in BHK-21 cells.** BHK-21 c15 cells were infected with DENV-2 and then treated with either torin-1 (1 μM), spautin-1 (10 μM), or 1% DMSO as a control. Samples were collected 48 hpi for titration. Data was analyzed by one-way ANOVA with a Dunnett's multiple comparisons test. (*) p<0.05, (**) p<0.01, (***) p<0.001.

arbovirus interface is primarily informed by one host, mammals [22]. It is known that virus infections can induce an autophagic response in Diptera, specifically *Drosophila*; however, there are conflicting reports as to the role of autophagy during infection [30–33]. Aside from having an intact and functional autophagy pathway, little is known about autophagy in mosquitoes. In an attempt to close this knowledge gap, this study characterized the autophagic response to arbovirus infection in the tractable Aag2 *Aedes aegypti* cell culture system. Our findings suggest that while autophagy is induced, it seems to have a limited role during arbovirus infection of mosquito cell culture. Further, we demonstrate that while commonly used

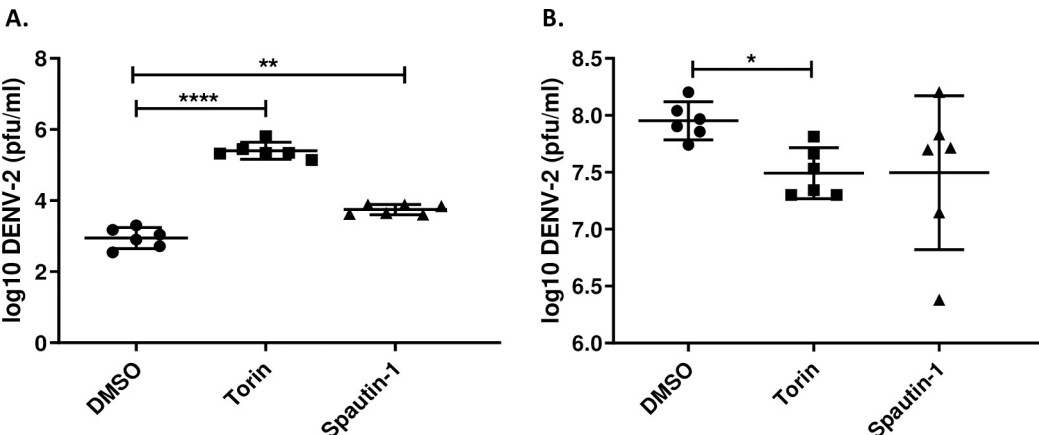

**Fig 7. Pharmacological modulators of autophagy increase DENV-2 titers in Aag2 cells, but not C6/36 cells.** Both A) Aag2 and B) *Aedes albopictus* C6/36 cells were infected with DENV-2 and then treated with either torin-1 (1 μM), spautin-1 (10 μM), or 1% DMSO as a control. Samples were collected 48 hpi for titration. Data was analyzed by one-way ANOVA with a Dunnett's multiple comparisons test. (*) p<0.05, (**) p<0.01, (***) p<0.001.

chemicals can modulate the autophagy pathway in mosquito cells, their effects on viruses are largely cell line dependent.

In agreement with a previous report, we found spautin-1 treatment significantly reduced the release of infectious DENV-2 particles during infection of mammalian cells [21]. In addition, Mateo et al. also observed that levels of extracellular viral RNA remained unchanged compared to the non-spautin-1 treated controls and that DENV-2 particles recovered from spautin-1 treated cells retained their 'pr' peptide after exocytosis [21]. During assembly, the viral nucleocapsid composed of genomic RNA and capsid protein, is packaged into a host-cell-derived envelope containing 180 copies of each of the E and prM structural proteins. Subsequently, immature virions undergo maturation in the trans-golgi network whereby the 'pr' peptide is proteolytically removed from M by host furin [34]. 'pr' remains in association with the virion through exocytosis after which it disassociates. Improper cleavage or disassociation can result in the production of immature particles which have been shown to be less infectious than the fully mature virions [35]. Based on this evidence it was hypothesized that either retention of the 'pr' peptide or altered virion confirmation diminished DENV-2 infectiousness after spautin-1 treatment [21]. However, this outcome appears to be limited to mammalian cells as we observed increased infectiousness after spautin-1 treatment of infected Aag2 mosquito cells. The reasons for this are unknown. A previous study demonstrated that furin substrate specificity differs between vertebrate and invertebrate cells, which may explain why there is a higher proportion of immature virions produced in mosquito cells compared to mammalian cells [36]. Interestingly, there doesn't appear to be a reduction in infectiousness. One potential explanation could be that DENV adopts different conformational arrangements in mosquito and mammalian cells. In fact, at 28°C, the temperature at which mosquito cells are cultured, mature DENV virions contain 90 E-glycoprotein homodimers arranged in a tightly packed antiparallel formation with a smooth appearance [37]. Conversely, at 37°C, E-glycoprotein homodimers adopt a very different conformation in which virions appear bumpy with large gaps allowing visualization of the underlying lipid membrane bilayer [37]. It is possible that 'pr' cleavage and disassociation are not required to form infectious virions at 28°C due to the observed conformational rearrangements and therefore, spautin-1 has no adverse effects and potentially beneficial effects on the production of infectious virus particles. This could explain the differences observed between mosquito cells and mammalian cells upon spautin-1 treatment.

Paradoxically, chemical induction and suppression of autophagy and inhibition of autophagolysosomal acidification all resulted in increased DENV-2, ZIKV and CHIKV titers in Aag2 cells at the assayed time points. In an attempt to reconcile these conflicting results, we performed RNAi suppression of autophagy genes. For both DENV-2 and CHIKV, no effects were observed upon suppression of Atg5, Atg14 or Atg8 despite high levels of suppression. Such results suggest that autophagy plays a limited role in mediating arbovirus infection of mosquito cells and that the effects conferred by the chemicals may be indirectly beneficial. Alternatively, it could suggest that >80% suppression was not sufficient to diminish autophagic activity and therefore, the RNAi suppression infection assays were not sensitive enough to identify a role for autophagy during arbovirus infection. A third possible explanation could be that mosquitoes encode functional redundancy in key autophagic genes. It is known that Atg8 is part of a larger family of homologs that include genes such as GABARAP and GABARAPL. Recent work has shown that autophagy remains functional upon deletion of any of these homologs individually [38]. Further, humans encode four isoforms of Atg4 and that elimination of all four is necessary to eliminate cleavage of pro-LC3/GABARAP prior to lipidation [39]. While additional homologs and/ or isoforms of Atg5 and Atg14 have not been identified in other systems, functional redundancy may exist in mosquitoes.

In this study we observed a significant increase in viral titers upon spautin-1 and torin-1 treatment in Aag2 cells; however, we observed no effect in another mosquito cell line, C6/36 cells. While both are mosquito cell lines, there are significant differences that could affect the chemical-arbovirus-cell interface. Aag2 cells are derived from *Ae. aegypti* mosquitoes whereas C6/36 cells originate from *Ae. albopictus* mosquitoes; however, due to the evolutionary conservation of autophagy it is unlikely that there are significant species-level differences in autophagic activity. A potentially more important factor could be the tissue source of the cells [40]. Mosquito cell culture is generally initiated by grinding up larvae (C6/36 cells) or embryos (Aag2 cells) and recovering cells amenable to growth in culture. Consequently, the specific cell type of the cells in culture is unknown. It could be that the differences arise because they are derived from different tissues and therefore respond differently to DENV-2 in the presence of chemicals. In addition, previous reports have demonstrated that Aag2 cells are persistently infected with multiple insect-specific viruses while C6/36 cells are not [41]. It is possible that additional viruses in the system could alter how the cells respond to both chemicals and arboviruses. Finally, the primary antiviral immune response, RNAi, is functionally impaired in C6/36 cells [27]. How this would impact the host cells' response to chemicals targeting the autophagy pathway during viral infection is unclear; however, previous studies have demonstrated that autophagy can selectively degrade components of the RNAi pathway [42, 43]. If RNAi is actively targeting viruses during infection, than induction with torin-1 could lead to increased viral replication in Aag2 cells but not C6/36 cells. Conversely, suppression of autophagy with spautin-1 would in theory increase available RNAi machinery thereby enabling a more potent RNAi response and reduction of viral titers; however, this is not what was observed. This may reflect the fact that RNAi genes are not significantly upregulated during arbovirus infection so suppression of autophagy has little effect on the normal levels of RNAi while increased autophagic activity could reduce activity allowing the viruses to replicate to higher titers [44]. Further research will be needed to fully elucidate autophagy-RNAi interactions and how they impact arbovirus infection of mosquitoes.

Comparing our results to what has been reported in mammalian systems reveals that at least for DENV-2 and ZIKV the general patterns are consistent. The increase in titers upon TOR-inhibition with torin-1 suggests that both DENV-2 and ZIKV benefit from downstream pathways regulated by TOR, one of which is autophagy. We observed increased flavivirus titers in Aag2 cells upon torin-1 treatment suggesting that flaviviruses benefit from an autophagic state and this is consistent with what has been found in mammalian cells [9, 21]. However, increased titers were also observed upon suppression of autophagy by spautin-1 suggesting an antiviral role. While these outcomes seem contradictory, they are consistent with the current model for DENV-autophagy interactions. In mammalian cells, DENV and DENV replication complexes are targeted for degradation by autophagy while simultaneously benefitting from more efficient mature virion processing and increased energy availability as a result of virus-induced lipophagy [9, 19–21]. While the flavivirus data was consistent with what has been observed in mammalian systems, the CHIKV data was not. Numerous studies have found that autophagy functions in an antiviral manner during infection with members of the family *Togaviridae*, specifically alphaviruses [10, 13, 15, 17]. While inhibition of autophagy did result in increased CHIKV titers as would be expected, induction also increased titers. This discrepancy may reflect a difference in the role of autophagy during CHIKV infection of mosquito and mammalian cells.

This study provides an important first-step in understanding autophagy-arbovirus interactions within mosquitoes and highlights differences between the mammalian and mosquito autophagic response to infection. However, future work assessing these interactions within mosquitoes will be needed due to the limitations of our cell culture system. As previously

mentioned, the origin of mosquito cell culture is unknown and therefore, it is difficult to determine if, in the context of natural infection, virus-autophagy interactions may function differently. Further, during infection, arboviruses must infect and replicate in a diversity of tissues and cell types before being transmitted. Previous studies have shown that autophagy can function differently between cell types and tissue types, and it is therefore possible that during the course of mosquito infection autophagy might interact differently with viruses depending on the tissue and cell type [40]. In addition, a recent study highlighted the genomic differences observed between C6/36 *Aedes albopictus* cells and *Aedes albopictus* mosquitoes [45]. The cells used in this study have been in culture for decades and may not accurately recapitulate the cellular environment of *Aedes aegypti*. Regardless, this study better defines autophagy-arbovirus interactions in a medically relevant vector system and provides insights that will be important for future *in vivo* studies.

## Supporting information

**S1 Supporting Information. Contains the materials and methods associated with supplemental figures.**
(DOCX)

**S1 Fig. Chemical modulators of autophagy have minimal toxicity in Aag2 cells.** Aag2 cells were treated with either torin-1 (1 μM), spautin-1 (10 μM), bafilomycin A1 (1 μM), 1% DMSO, or untreated for 24 hrs (n = 4). Subsequently, the MTT cell viability assay was performed to determine if the chemicals had cytotoxic effects. Data was analyzed by one-way ANOVA with a Dunnett's multiple comparisons test. Red data points were determined to be outliers by Grubb's test for outliers and were not included in the analysis.
(TIF)

**S2 Fig. Zika virus induces autophagy in *Aedes aegypti* Aag2 cells.** A) Representative immunoblot of Aag2 lysates upon ZIKV infection (M.O.I. 0.5) and/ or chemical treatment (1% DMSO, 1 μM bafilomycin A1, 1 μM torin-1 or 10 μM spautin-1) 48 hpi. B) Fold-change in the ratio of Atg8-PE to β-actin band intensities as determined by ImageJ. Includes data from four experimental replicates. C) Representative confocal microscopy images of Atg8-EGFP expressing Aag2 cells ± ZIKV infection (M.O.I. 0.1) and ±1 μM bafilomycin A1 24 hpi. Blue (nuclei), green (Atg8-EGFP + puncta). D) The number of Atg8+ puncta were quantified using the ImageJ Puncta Analyzer plug-in from ~50 Atg8-EGFP expressing Aag2 cells ± ZIKV infection (M. O.I. 0.1) and ±1 μM bafilomycin-A1 24 hpi. Combined data from three blinded experimental replicates. Data were analyzed by One-way ANOVA with a Sidak's multiple comparisons test. (*) $p < 0.05$, (**) $p < 0.01$, (***) $p < 0.001$.
(TIF)

**S3 Fig. Both induction and inhibition of autophagy increase ZIKV titers in Aag2 cells.** *Aedes aegypti* Aag2 cells were infected with ZIKV followed by chemical treatment (1% DMSO, 1 μM bafilomycin A1, 1 μM torin-1 or 10 μM spautin-1) or treated with dsRNA against Atg5, Atg14, or non-specific control luciferase genes two days prior to infection with ZIKV. Samples were collected for titration 48 hpi. Data was analyzed by one-way ANOVA with a Dunnett's multiple comparisons test. (*) $p < 0.05$, (**) $p < 0.01$, (***) $p < 0.001$.
(TIF)

**S4 Fig. Efficient silencing of autophagy genes in mosquito cells.** Aag2 cells were treated with dsRNA targeting Atg5, Atg14, or Atg8 and assayed for suppression 48 hours post transfection. Silencing efficiency of A) Atg5 and B) Atg14 was determined by $\Delta\Delta C_T$ analysis with luciferase

samples as the non-targeting control group and GAPDH as a reference gene. Data was analyzed with a two-tailed t-test. C) Silencing efficiency of Atg8 was determined by immunoblot. (TIF)

## Acknowledgments

We would like to thank Dr. Gregory Ebel and Benjamin Dodd for their thoughtful insights and early support of this work.

## Author Contributions

**Conceptualization:** Doug E. Brackney.

**Data curation:** Doug E. Brackney, Maria A. Correa, Duncan W. Cozens.

**Funding acquisition:** Doug E. Brackney.

**Investigation:** Maria A. Correa, Duncan W. Cozens.

**Methodology:** Doug E. Brackney, Maria A. Correa, Duncan W. Cozens.

**Supervision:** Doug E. Brackney.

**Writing – original draft:** Doug E. Brackney.

**Writing – review & editing:** Doug E. Brackney, Maria A. Correa.

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
