## [Decision Letter · Decision Letter 0]

1 Nov 2019

Dear Dr Brackney:

Thank you very much for submitting your manuscript "A limited role for autophagy during arbovirus infection of mosquito cells" (#PNTD-D-19-01476) for review by PLOS Neglected Tropical Diseases. Your manuscript was fully evaluated at the editorial level and by independent peer reviewers. The reviewers appreciated the attention to an important problem, but raised some substantial concerns about the manuscript as it currently stands. These issues must be addressed before we would be willing to consider a revised version of your study. We cannot, of course, promise publication at that time.

We therefore ask you to modify the manuscript according to the review recommendations before we can consider your manuscript for acceptance. Your revisions should address the specific points made by each reviewer. 

When you are ready to resubmit, please be prepared to upload the following:

(1) A letter containing a detailed list of your responses to the review comments and a description of the changes you have made in the manuscript.

(2) Two versions of the manuscript: one with either highlights or tracked changes denoting where the text has been changed (uploaded as a "Revised Article with Changes Highlighted" file); the other a clean version (uploaded as the article file).

(3) If available, a striking still image (a new image if one is available or an existing one from within your manuscript). If your manuscript is accepted for publication, this image may be featured on our website. Images should ideally be high resolution, eye-catching, single panel images; where one is available, please use 'add file' at the time of resubmission and select 'striking image' as the file type. 

Please provide a short caption, including credits, uploaded as a separate "Other" file. If your image is from someone other than yourself, please ensure that the artist has read and agreed to the terms and conditions of the Creative Commons Attribution License at http://journals.plos.org/plosntds/s/content-license (NOTE: we cannot publish copyrighted images). 

(4) If applicable, we encourage you to add a list of accession numbers/ID numbers for genes and proteins mentioned in the text (these should be listed as a paragraph at the end of the manuscript). You can supply accession numbers for any database, so long as the database is publicly accessible and stable. Examples include LocusLink and SwissProt.

(5) To enhance the reproducibility of your results, we recommend that you deposit your laboratory protocols in protocols.io, where a protocol can be assigned its own identifier (DOI) such that it can be cited independently in the future. For instructions see http://journals.plos.org/plosntds/s/submission-guidelines#loc-methods

While revising your submission, please upload your figure files to the Preflight Analysis and Conversion Engine (PACE) digital diagnostic tool, https://pacev2.apexcovantage.com/ PACE helps ensure that figures meet PLOS requirements. To use PACE, you must first register as a user. Then, login and navigate to the UPLOAD tab, where you will find detailed instructions on how to use the tool. If you encounter any issues or have any questions when using PACE, please email us at figures@plos.org.

We hope to receive your revised manuscript by Dec 31 2019 11:59PM. If you anticipate any delay in its return, we ask that you let us know the expected resubmission date by replying to this email.

To submit a revision, go to https://www.editorialmanager.com/pntd/ and log in as an Author. You will see a menu item call Submission Needing Revision. You will find your submission record there. 

Sincerely,

Rhoel Ramos Dinglasan

Associate Editor

Paulo Pimenta

Deputy Editor

Reviewer's Responses to Questions

**Key Review Criteria Required for Acceptance?**

**Methods**

-Are the objectives of the study clearly articulated with a clear testable hypothesis stated?

-Is the study design appropriate to address the stated objectives?

-Is the population clearly described and appropriate for the hypothesis being tested?

-Is the sample size sufficient to ensure adequate power to address the hypothesis being tested?

-Were correct statistical analysis used to support conclusions?

-Are there concerns about ethical or regulatory requirements being met?

Reviewer #1: Yes - 

The authors do not have a clear hypothesis due to the nature of the study. However the question that is being addressed on the Role of autophagy in Dengue/Zika virus life cycle in insect cells is of significance.

Reviewer #2: Specific comments are presented in a separate file. The rationality for multiple methods is not clear, nor are the timepoints chosen, or why some experiments were completed for some viruses but not other viruses or chemical modulators.

Reviewer #3: (No Response)

**Results**

-Does the analysis presented match the analysis plan?

-Are the results clearly and completely presented?

-Are the figures (Tables, Images) of sufficient quality for clarity?

Reviewer #1: Yes.

Reviewer #2: Specific comments are presented in a separate file. I believe the results could be presented in a clearer and more concise manner (perhaps a table summarizing the results would be useful base on the large number of separate and combined treatments). Figures 1-3 are not of sufficient quality for reading.

Reviewer #3: (No Response)

**Conclusions**

-Are the conclusions supported by the data presented?

-Are the limitations of analysis clearly described?

-Do the authors discuss how these data can be helpful to advance our understanding of the topic under study?

-Is public health relevance addressed?

Reviewer #1: Appropriate conclusions have been drawn from the data acquired. However, some more experiments might need to be performed to arrive at a conclusion of the involvement of autophagy in flavi infection of infected cells.

Reviewer #2: Specific comments are presented in a separate file. The results and the discussion bring up separate points. There needs to be an improved flow between the sections to tie them together.

Reviewer #3: (No Response)

**Editorial and Data Presentation Modifications?**

Reviewer #1: (No Response)

Reviewer #2: Specific comments are presented in a separate file. Please be consistent with including timepoints of relevance, as well as DENV serotype and the use of hyphens.

Reviewer #3: (No Response)

**Summary and General Comments**

Reviewer #1: In this paper the authors have described the influence of autophagy in Dengue and Zika virus production in insect cells. Autophagy has been described to play a critical role in flaviviral lifecycle as well as in the innate immune responses in human host cells. Both these viruses are however mosquito-borne and the corresponding role of autophagy in insect cells have not been investigated in the past. It is therefore certainly a significant knowledge gap that currently exists in the transmission cycle of these viruses. The findings presented in this paper are relevant; however, at the current state they seem rather preliminary and inconclusive. The authors need to clarify a few additional points via experiments and textual changes:

Specific comments:

Line 10: The process of Dengue induced lipophagy and its role in processing mature viral progenies was actually demonstrated in this paper (Zhang et al., 2018); this needs to be cited. Along the same lines, since both Dengue and Zika have been reported to induce autophagy for the purpose of hydrolyzing lipid droplets, it might be relevant to measure their abundance in infected insect cells, via simple Nile Red staining. This will provide useful information on similarities and differences between human and insect cells.

Figure 1-3: The authors have quantitated the appearance of EGFP-Atg8 punctae upon transient transfection of the plasmid into cells. From the images shown, it appears that the transfection efficiency is extremely low in these cells – hence the quantitative estimates might not be reflective of the whole population. It might be a lot more interpretable if they stably expressed EGFP-Atg8 in these cells. 

Figure 4: What is the authors’ explanation for the increase in viral titers with both inhibition and induction of autophagy? Also a more systematic time course analysis is probably called for to analyse how autophagy contributes to the early versus late stages in the viral life cycle.

Figure 5: Depletion of the three autophagosomal genes needs to be confirmed by immunoblotting at protein expression levels. If the extent of depletion is low, it is possible that the lack of a significant effect on viral titers is because of residual activity of the genes. Also, how do the authors reconcile the discrepancy in the phenotype with genetic depletion versus the pharmacological inhibition experiments? Gene-depletions show non-significant effect on viral titers whereas induction or inhibition with pharmacological compounds clearly increases viral titers.

Lastly, a lot of the conclusion from each of the figures has been compiled together with the discussion section. It will be a lot easier for the reader if the conclusion of the authors for each figure was added to the results section, while its implications and interpretations discussed in the discussion section.

Minor comments:

Figure S1A: The annotation of Atg8-PE and Atg8 are reversed – Atg8-PE runs as a smaller molecular weight species on the gel

There are several typographical errors in the manuscript that needs to be corrected.

Zhang, J., Lan, Y., Li, M.Y., Lamers, M.M., Fusade-Boyer, M., Klemm, E., Thiele, C., Ashour, J., Sanyal, S., 2018. Flaviviruses Exploit the Lipid Droplet Protein AUP1 to Trigger Lipophagy and Drive Virus Production. Cell Host Microbe 23, 819–831.e5.

Reviewer #2: Specific comments are presented in a separate file. I think the study is interesting overall, but the manuscript as written would greatly benefit from increased explanations, better flow and clarity throughout. Please either complete the experiments necessary to make the study fully factorial, or provide an explanation for why not all experiments were completed (across viruses, chemical modulators and timepoints).

Reviewer #3: (No Response)

PLOS authors have the option to publish the peer review history of their article (what does this mean?). If published, this will include your full peer review and any attached files.

Reviewer #1: Yes: Sumana Sanyal

Reviewer #2: No

Reviewer #3: No

---

## [Decision Letter · Decision Letter 1]

18 Feb 2020

Dear Dr Brackney,

Thank you very much for submitting your manuscript "The impact of autophagy during arbovirus infection of mosquito cells" for consideration at PLOS Neglected Tropical Diseases. As with all papers reviewed by the journal, your manuscript was once again reviewed by members of the editorial board and by two independent reviewers. 

Although we remain appreciative of the topic, the last revised version still needs to be further improved. To consider the manuscript for publication, we encourage you to use the comments and suggestions of the reviewers from the first and second rounds of review to modify the manuscript, with a goal of increasing the clarity and transparency of the results.

Sincerely,

Rhoel Ramos Dinglasan

Associate Editor

Paulo Pimenta

Deputy Editor

Reviewer's Responses to Questions

**Key Review Criteria Required for Acceptance?**

**Methods**

-Are the objectives of the study clearly articulated with a clear testable hypothesis stated?

-Is the study design appropriate to address the stated objectives?

-Is the population clearly described and appropriate for the hypothesis being tested?

-Is the sample size sufficient to ensure adequate power to address the hypothesis being tested?

-Were correct statistical analysis used to support conclusions?

-Are there concerns about ethical or regulatory requirements being met?

Reviewer #2: More in-depth descriptions are provided in the Reviewer Comments file. Overall the methods were straight forward and clear.

Reviewer #3: (No Response)

**Results**

-Does the analysis presented match the analysis plan?

-Are the results clearly and completely presented?

-Are the figures (Tables, Images) of sufficient quality for clarity?

Reviewer #2: There needs to be some work in the results section, particularly with the figures to increase clarification, espescially with regards to significance asterisks and blotting images.

Reviewer #3: (No Response)

**Conclusions**

-Are the conclusions supported by the data presented?

-Are the limitations of analysis clearly described?

-Do the authors discuss how these data can be helpful to advance our understanding of the topic under study?

-Is public health relevance addressed?

Reviewer #2: The conclusions are supported by the data presented, yes. I think there could be a larger tie in to genes reported in mosquito RNAseq studies. The data is paradoxical, but these issues are addressed and discussed accordingly. As this is a study with DENV-2 and Aag2 cells, there is an implied public health relevance, yes.

Reviewer #3: (No Response)

**Editorial and Data Presentation Modifications?**

Reviewer #2: Please see attached reviewer comments.

Reviewer #3: (No Response)

**Summary and General Comments**

Reviewer #2: Please see attached reviewer comments.

Reviewer #3: Most of the important points I raised have been addressed. However, I wish the authors would have been more open to the reviewers comments. They are meant to improve the manuscript and should be seriously considered, rather than answered lightly and with general comments such as "this is science". 

I do not understand the reluctance to implement changes that where highlighted by both reviewers, such as a better description of time points throughout the manuscript, that overall stil lacks precision. 

Adding a small scheme that recapitulates the autophagy pathway, and highlighting the steps that are affected by the various drugs used in the paper would be very welcome and does not necessitate much extra work. Explaining in a more comprehensive way what "autophagic flux" is in the manuscript, as has been done in the answers to reviewers would be helpful to readers. 

Line numbers were not always referred to properly in the reply to reviewers (where is Atg8-PE defined ? It is not page 5 line 106 as written) and leave a general impression of sloppiness.

PLOS authors have the option to publish the peer review history of their article (what does this mean?). If published, this will include your full peer review and any attached files.

Reviewer #2: No

Reviewer #3: No
---

## [Decision Letter · Decision Letter 2]

26 Mar 2020

Dear Dr Brackney,

We are pleased to inform you that your manuscript 'The impact of autophagy during arbovirus infection of mosquito cells' has been provisionally accepted for publication in PLOS Neglected Tropical Diseases.

Before your manuscript can be formally accepted you will need to complete some formatting changes, which you will receive in a follow up email. A member of our team will be in touch with a set of requests. Also, please consider carefully additional comments by one reviewer in the hope of further improving the manuscript as it progresses forward to publication.

Best regards,

Rhoel Ramos Dinglasan

Associate Editor

Paulo Pimenta

Deputy Editor

Reviewer's Responses to Questions

**Key Review Criteria Required for Acceptance?**

**Methods**

-Are the objectives of the study clearly articulated with a clear testable hypothesis stated?

-Is the study design appropriate to address the stated objectives?

-Is the population clearly described and appropriate for the hypothesis being tested?

-Is the sample size sufficient to ensure adequate power to address the hypothesis being tested?

-Were correct statistical analysis used to support conclusions?

-Are there concerns about ethical or regulatory requirements being met?

Reviewer #2: Methods are sound.

Reviewer #3: Methods are appropriately described.

**Results**

-Does the analysis presented match the analysis plan?

-Are the results clearly and completely presented?

-Are the figures (Tables, Images) of sufficient quality for clarity?

Reviewer #2: Results are sound.

Reviewer #3: Results are clear well presented.

**Conclusions**

-Are the conclusions supported by the data presented?

-Are the limitations of analysis clearly described?

-Do the authors discuss how these data can be helpful to advance our understanding of the topic under study?

-Is public health relevance addressed?

Reviewer #2: Conclusions are sound.

Reviewer #3: Conclusions are supported by the data.

**Editorial and Data Presentation Modifications?**

Reviewer #2: Minor modifications noted in the attached comments.

Reviewer #3: (No Response)

**Summary and General Comments**

Reviewer #2: (No Response)

Reviewer #3: I am satisfied with the revisions and recommend the manuscript for publication.

PLOS authors have the option to publish the peer review history of their article (what does this mean?). If published, this will include your full peer review and any attached files.

Reviewer #2: No

Reviewer #3: No
